# On the Therapeutic Potential of ERK4 in Triple-Negative Breast Cancer

**DOI:** 10.3390/cancers15010025

**Published:** 2022-12-21

**Authors:** Fadia Boudghene-Stambouli, Mathilde Soulez, Natalia Ronkina, Anneke Dörrie, Alexey Kotlyarov, Ole-Morten Seternes, Matthias Gaestel, Sylvain Meloche

**Affiliations:** 1Institute for Research in Immunology and Cancer, Montreal, QC H3T 1J4, Canada; 2Department of Pharmacology and Physiology, Université de Montréal, Montreal, QC H3C 3J7, Canada; 3Institute of Cell Biochemistry, Hannover Medical School, D-30625 Hannover, Germany; 4Department of Pharmacy, UiT The Arctic University of Norway, 9037 Tromso, Norway; 5Molecular Biology Program, Faculty of Medicine, Université de Montréal, Montreal, QC H3C 3J7, Canada

**Keywords:** MAP kinase, ERK4, AKT, breast cancer, cell lines

## Abstract

**Simple Summary:**

Patients with triple-negative breast cancer have a poor outcome owing to the clinically aggressive behavior of the disease and the lack of hormonal or targeted therapies. Identification of new actionable targets to guide the development of effective treatments remains a critical clinical need. It has been recently suggested that the protein kinase ERK4 has oncogenic signaling activity in triple-negative breast cancer cells and represents a promising novel therapeutic target. However, we raise questions about the experimental approaches used to validate the oncogenic function of ERK4 in breast cancer.

**Abstract:**

ERK3 and ERK4 define a distinct and understudied subfamily of mitogen-activated protein kinases (MAPKs). Little is known about the physiological roles of these atypical MAPKs and their association with human diseases. Interestingly, accumulating evidence points towards a role for ERK3 and ERK4 signaling in the initiation and progression of various types of cancer. Notably, a recent study reported that ERK4 is expressed in a subset of triple-negative breast cancer (TNBC) cell lines and that this expression is critical for AKT activation and for sustaining TNBC cell proliferation in vitro and tumor growth in mice. The authors also showed that depletion of ERK4 sensitizes TNBC cells to phosphatidylinositol-3-kinase (PI3K) inhibitors. They concluded that ERK4 is a promising therapeutic target for TNBC and has potential for combination therapy with PI3K inhibitors. Here, we raise concerns about the cellular models and experimental approaches used in this study, which compromise the conclusions on the oncogenic role of ERK4 in TNBC.

## 1. Introduction

ERK3/MAPK6 (*MAPK6* gene) along with its paralog ERK4/MAPK4 (*MAPK4* gene) define a distinct subfamily of atypical MAPKs [1,2]. Much remains to be learned about the substrates and biological functions of these signaling enzymes. ERK3 is expressed ubiquitously in adult tissues, while ERK4 has a limited expression profile and is mainly detected in the central nervous system [3,4,5,6]. Contrary to classical MAPKs such as ERK1/2 that phosphorylate a large number of substrates [7,8], ERK3 and ERK4 appear to have a more restricted substrate specificity. Their only reported substrate is the protein kinase MK5 [9,10,11,12,13,14]. Genetic invalidation studies have revealed that mice lacking ERK3 kinase activity or expression are born at normal mendelian frequency and show no overt abnormalities [15,16]. Mice with genetic disruption of ERK4 are also viable and fertile with no apparent phenotype [5]. Interestingly, recent work suggests that ERK3 and ERK4 signaling may play a role with cancer initiation and progression. Specifically, it has been proposed that ERK4 promotes cancer progression via the non-canonical activation of AKT/mTOR signaling [17,18,19,20]. In order to investigate the role of ERK4 in TNBC, Wang et al. [21] surveyed a panel of TNBC cell lines and found that ERK4 is highly expressed in MDA-MB-231, Hs578T and HCC1937 cell lines (referred to as MAPK4-high), while other TNBC cell lines express low-to-undetectable levels of the kinase. They used the NSCLC cell line H1299 as a positive control of ERK4 expression. The MAPK4-high cell lines were then used throughout the study to validate the oncogenic role of ERK4. Compared to ERK3, which is expressed in most tissues and cell lines, ERK4 shows a restricted expression profile and is generally found at low or undetectable levels in most cell lines [6]. Previous independent observations from our laboratories suggested that ERK4 is not expressed in some of the MAPK4-high cell lines described in the Wang et al. [21] paper, prompting us to examine in more detail the expression of ERK4 in their cell line panel. 

## 2. Results and Discussion

We first measured the abundance of *MAPK4* mRNA by qRT-PCR. We found that expression of *MAPK4* gene was undetectable in Hs578T and very low-to-undetectable (Ct > 35) in MDA-MB-231 and HCC1937 cells, but was reproducibly detected in HeLa and HEK 293T cells (Figure 1A). No expression of *MAPK4* mRNA could be measured in H1299 cells. In contrast, *MAPK6* expression was detected in all cell lines (Figure 1A). Examination of internal transcriptomic data obtained by RNA-seq analysis of MDA-MB-231 and Hs578T cells also revealed that *MAPK4* mRNA is undetectable in these cells, while *MAPK6* is expressed at significant levels (Figure 1B). These data indicate that the *MAPK4* gene is expressed at very low-to-undetectable levels in the MAPK4-high cell lines of Wang et al. [21].

We next measured the expression of the ERK4 protein in TNBC cell lines. Two different antibodies were used for these experiments: the anti-MAPK4 from Abcepta (AP7298b) used by Wang et al. [21] and a validated custom polyclonal ERK4 antibody. We failed to detect a specific ERK4 band in any of the TNBC cell lines, including Hs578T cells transfected with human ERK4 cDNA, using the anti-ERK4 antibody from Abcepta (Figure 1C,D and Appendix A). Using our custom anti-ERK4 antibody, we observed an immunoreactive band of ~70 kDa in control HeLa cells and HEK 293T cells (Figure 1C and Appendix A). This band was eliminated by siRNA-mediated depletion of ERK4 in HeLa cells. In agreement with the mRNA expression data, no expression of ERK4 protein was detected in MDA-MB-231, Hs578T and HCC1937 cell lines, or in control H1299 NSCLC cells (Figure 1C and Appendix A). In contrast, the ERK3 protein was expressed to detectable levels in all cell lines (Figure 1E).

The Cancer Cell Line Encyclopedia (CCLE) is a trusted database that compiles gene expression data for ~1000 cell lines available from public cell line repositories [22]. Interrogation of the CCLE database showed that the *MAPK4* gene is expressed at very low or undetectable levels (TPM ≤ 1) in the TNBC cell lines and control H1299 cells (Figure 1F). In contrast, *MAPK6* was expressed in all TNBC cell lines. There was no correlation between the expression of *MAPK4* mRNA in CCLE TNBC cell lines and the expression of the ERK4 protein in the TNBC cell lines of Wang et al. [21]. Taken together, these findings are inconsistent with a significant expression and oncogenic function of ERK4 in the TNBC cellular models used by Wang et al. [21]. In further support of the lack of role for ERK4, interrogation of the DepMap portal [23] indicates that CRISPR/Cas9-mediated depletion of ERK4 in MDA-MB-232, Hs578T and HCC1937 has no effect on the proliferation of these cell lines, consistent with the lack of expression of the *MAPK4* gene (Figure 1G).

Another concern raised by the Wang et al. [21] study relates to the proposed non-canonical activation of AKT by ERK4, which serves as the rationale for the author’s hypothesis that ERK4 depletion sensitizes TNBC cells to PI3K inhibitors. ERK4 is a member of the MAPK family of Ser/Thr kinases, which are proline-directed kinases [24,25,26]. Accordingly, ERK4 phosphorylates its only known substrate MK5 on Thr182, which is followed by a Pro residue [9,10]. The activation loop threonine of AKT1, AKT2 and AKT3 (Thr308 in AKT1) is not followed by a proline, raising questions about the validity and significance of this mechanism. We have examined the ability of ERK4 to phosphorylate AKT on Thr308 and Ser473 in Hs578T and HEK 293T cells. We found that overexpression of either ERK4 or catalytically-inactive ERK4 KK49/50AA had no significant impact on the phosphorylation of AKT on Thr308 or Ser473 (Figure 2 and Appendix A). In contrast, the expression of ERK4, but not ERK4 KK49/50AA clearly induced the phosphorylation of MK5 on Thr182 and caused MK5-mediated phosphorylation of ERK4, which is evident by the retarded migration of ERK4 band [27].

In conclusion, our findings do not support the claim that ERK4 promotes TNBC growth and is a promising therapeutic target for this cancer [21]. We found no evidence that ERK4 is expressed to significant levels in the MAPK4-high cell lines used by the authors to validate the oncogenic role of ERK4. We also question the significance and generalization of the non-canonical activation of the AKT/mTOR pathway by ERK4 proposed by these authors. Our observations do not call into question the possible role of ERK4 in other solid cancers, which was not evaluated in this study.

## 3. Materials and Methods

### 3.1. Cell Lines and Cell Culture

All cell lines were obtained from the American Type Culture Collection. The TNBC cell lines MDA-MB-231, Hs578T and HCC1937 were further authenticated by short tandem repeat profiling (ATCC). MDA-MB-231, Hs578T and HEK 293T cells were cultured in DMEM supplemented with 10% fetal bovine serum (FBS) and antibiotics. H1299, HCC1937 and HeLa cells were cultured in RPMI supplemented with 10% FBS and antibiotics. MCF10A cells were cultured in DMEM/F12 supplemented with 10% FBS, epidermal growth factor, hydrocortisone, insulin and antibiotics. All cell lines were routinely tested for mycoplasma contamination.

### 3.2. Transfections and RNA Interference

For ectopic expression of ERK4, Hs578T cells grown in 60 mm culture plates were transfected with Lipofectamine 3000 and 3 µg of DNA of the indicated HA-ERK4 and Flag-MK5 plasmids. After 48 h, the cells were lysed in 100 µL of RIPA buffer and extracts were analyzed by Western blotting. For expression of GFP constructs, HEK 293T cells were transiently transfected with polyethylenimine and 1 µg of DNA per well in 12-well plates. After 24 h, the cells were lysed in the plate with 200 µL of 2X SDS-loading buffer per well. For siRNA treatment, 4 × 10^4^ HeLa cells were seeded into 12-well plates on the day before transfection. Next day, the cells were transfected with HiPerfect and 3 µL of 10 µM ERK4 siRNA from Santa Cruz Biotechnology (sc-62280) or control siRNA from Qiagen (1027280). After 48 h, the cells were lysed in the plate with 100 µL of 2X SDS-loading buffer.

### 3.3. Western Blot Analysis

Cell lysis and immunoblotting analysis were performed as described previously [28]. Custom polyclonal ERK4 antibody was raised in sheep against the full-length ERK4 protein produced in Sf9 cells and was affinity-purified on CH-Sepharose covalently coupled to the protein antigen [9,29]. Commercial antibodies for immunoblotting were obtained from the following suppliers: anti-ERK4 (1/1000; #AP7298b) from Abcepta; anti-ERK3 (1/1000; #ab53277) and anti-phospho-MK5(Thr182) (1/1000; #ab138668) from Abcam; anti-HA (1/5000; #901501) from BioLegend; anti-GAPDH (1/2000; #VMA00046), anti-HSC70 (1/2000; #sc-7298), and anti-GFP (1/1000; #sc-9996) from Santa Cruz Biotechnology; anti-phospho-AKT(Thr308) (1/1000; #13038) and anti-phospho-AKT(Ser473) (1/1000; #4051) from Cell Signaling Technology.

### 3.4. Real-Time Quantitative PCR Analysis

Total RNA was extracted with the RNeasy Mini Kit (Qiagen, Hilden, Germany) and reverse-transcribed with random primers and the High Capacity cDNA Archive Kit (Applied Biosystems, Foster City, CA, USA) as described by the manufacturer. Quantitative real-time PCR was performed as previously described [16]. The following forward (F) and reverse (R) primers were used: *MAPK4* (F: tacggggagaatgctctttg; R: ccggattacagggatggtc), *MAPK6* (F: gtacaagttgatccccgaaaat; R: caaaagcaggatcctccaga), *HPRT* (F: tgatagatccattcctatgactgtaga; R: caagacattctttccagttaaagttg), and *GAPDH* (F: agccacatcgctcagacac; R: gcccaatacgaccaaatcc).

### 3.5. Gene Expression Profiling

Total RNA was extracted from MDA-MB-231 and Hs578T cell lysates using an RNAeasy purification kit (Qiagen, Hilden, Germany). The quality of RNA was assessed on the Agilent 2100 Bioanalyzer and the RNA was quantified by QuBit fluorometry. Libraries (500 ng total RNA) were prepared using the KAPA Hyperprep messenger RNA (mRNA)-Seq Kit (KAPA Biosystems, Wilmington, MA, USA). Purified libraries were normalized by quantitative PCR (qPCR) using the KAPA Library Quantification Kit (KAPA Biosystems, Wilmington, MA, USA) and diluted to a final concentration of 10 nM. Sequencing was performed on the Illumina Nextseq500 using the Nextseq500 High Output Kit (75 cycles) (read length 1 × 80 bp). Around 24–33 M single-end pass-filter reads were generated per sample. Library preparation and sequencing was done at the Institute for Research in Immunology and Cancer Genomics core facility [30]. Alignment of RNA-seq reads was performed with the STAR aligner. The datasets generated during the current study are available from the corresponding author on reasonable request.

## Figures and Tables

**Figure 1 cancers-15-00025-f001:**
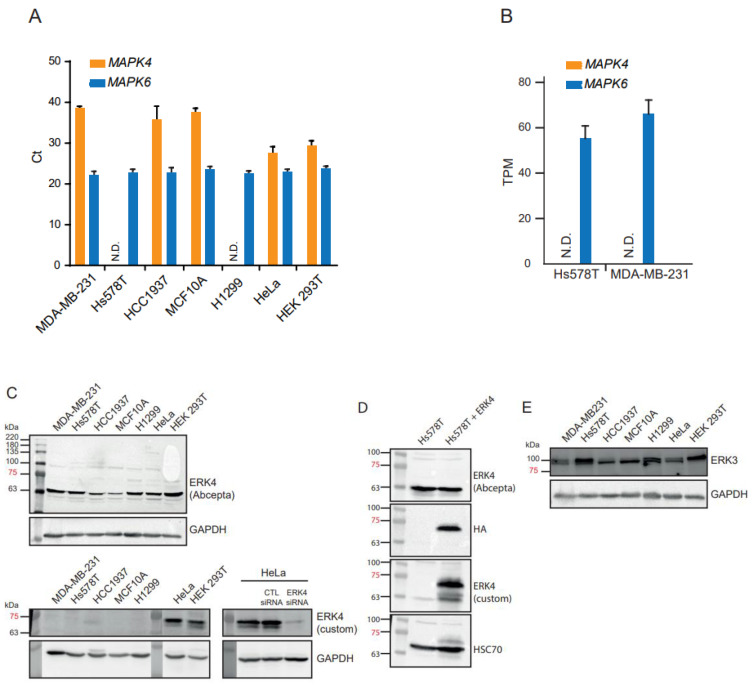
Expression of ERK4 mRNA and protein in TNBC cell lines. (**A**) Expression of *MAPK4* and *MAPK6* mRNA was measured by real-time qPCR in the human TNBC cell lines Hs578T, MDA-MB231 and HCC1937, the breast epithelial cell line MCF10A, and control H1299, HeLa and HEK 293T cells. Results are expressed as Ct. Data are means ± SD (*n* = 3). (**B**) Expression of *MAPK4* and *MAPK6* mRNA in Hs578T and MDA-MB-231 measured by RNA-seq. Results are expressed as TPM and correspond to the mean ± SD of triplicate samples. (**C)** Expression of ERK4 and ERK3 proteins were analyzed by Western blotting using the indicated antibodies. ERK4 was depleted from HeLa cells by siRNA. (**D**) Hs578T were transfected with HA-ERK4 and analyzed by Western blotting with the indicated antibodies. (**E**) Expression of the ERK3 protein was analyzed by Western blotting. (**F**) Expression of *MAPK4* and *MAPK6* mRNA in human TNBC cell lines and H1299 NSCLC cell line extracted from the CCLE database. (**G**) *MAPK4* gene dependency of breast cancer cell lines. The gene dependency score of *MAPK4* was estimated from a CRISPR/Cas9 loss-of-function screen of human cancer cell lines and extracted from the DepMap portal (https://depmap.org/portal/gene/MAPK4?tab=dependency; accessed on 19 October 2022). A cell line is considered dependent if the dependency score is <−0.5. A score of 0 indicates a gene that is not essential. The median scores of essential genes is −1.0.

**Figure 2 cancers-15-00025-f002:**
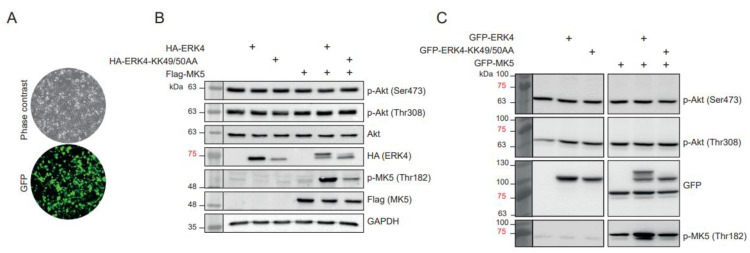
ERK4 catalytic activity has no impact on the phosphorylation of AKT on Thr308 and Ser473. (**A**) Hs578T TNBC cells were transfected with a GFP-expressing plasmid to show the high yield of transfection of the cells. (**B**) Hs578T cells were transfected with HA-ERK4 or catalytically-inactive HA-ERK4 KK49/50AA with or without Flag-MK5. Phosphorylation of endogenous AKT on Thr308 and Ser473 or of ectopically expressed Flag-MK5 was analyzed by immunoblotting with phospho-specific antibodies. Expression of HA-ERK4 was measured with anti-HA antibody. (**C**) HEK 293T cells were transfected with GFP-ERK4 or catalytically-inactive GFP-ERK4 KK49/50AA with or without GFP-MK5. Phosphorylation of endogenous AKT on Thr308 and Ser473 or of ectopically expressed GFP-MK5 was analyzed by immunoblotting with phospho-specific antibodies. Expression of GFP constructs was controlled with anti-GFP antibody.

## Data Availability

The datasets generated during the current study are available from the corresponding author on reasonable request.

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
