# Peer review of "On the Therapeutic Potential of ERK4 in Triple-Negative Breast Cancer"

_cancers, 2022, doi:10.3390/cancers15010025_

Round 1
Reviewer 1 Report
In this manuscript, the study raises concern about the results of the following study: Wang W, et al. MAPK4 promotes triple negative breast cancer growth and reduces tumor sensitivity to PI3K blockade. Nat Commun. 2022 Jan 11;13(1):245. doi: 10.1038/s41467-021-27921-1.
In the study conducted by Wang et al, they reported that high ERK4/MAPK4 expression defines a large subset of TNBC responsive to MAPK4 blockage and targeting it in this subset both represses tumor growth and sensitizes tumors to PI3K blockade. However, the results from this study are very different from Wang’s study in the following aspects.
- Concern about the expression level of MAPK4 in the cell lines. They showed that MAPK4 gene was expressed at very low-to-undetectable levels in the MAPK4-high cell lines of Wang et al. In agreement with the mRNA expression data, no expression of ERK4 protein was detected in MDA-MB-231, Hs578T and HCC1937 cell 64 lines, or in control H1299 NSCLC cells. The methods were the same as Wang’s and antibodies were also the same for protein detection. They further used another custom antibody to confirm their results. They also reviewed the CCLE database, which showed that MAPK4 gene is expressed at very low or undetectable levels (TPM ≤ 1) in the TNBC cell lines and control H1299 cells. The data from the study was very different from the original study “A comprehensive analysis of 8887 gene expression profiles in The Cancer Genome Atlas (TCGA) revealed that MAPK4 overexpression correlates with decreased overall survival, with particularly marked survival effects in patients with lung adenocarcinoma, bladder cancer, low-grade glioma, and thyroid carcinoma.” (Wang et al. PMID: 30688659).
2. Concern about the role of ERK4 in non-canonical activation of AKT, which serves as rationale for the author’s hypothesis that ERK4 depletion sensitizes TNBC cells to PI3K inhibitors. In contrast, this study proved that ERK4 had no significant impact on the phosphorylation of AKT so that not to activate AKT.
As the data from both study sound reasonably true, it will be difficult to judge which study is current and which is wrong. The best way to clean up the situation is to let the community to validate the results.
Author Response
We thank the reviewer for her/his comments.
Reviewer 2 Report
The authors in the manuscript entitled: "On the therapeutic potential of ERK4 in triple-negative breast cancer" raise concerns about the experimental approaches used in another study related to the role of ERK4 in TNBC.
Please prepare Figure B similarly to Figure A. So that you can compare the TPM in all tested cell lines.
I think it would be worthwhile to also present the data on gene expression (at the mRNA level) of MAPK4 and MAPK6 in triple-negative breast cancer contained in the TCGA database.
Commentary by Fadia Boudghene-Stambouli et al extends the discussion on the role of ERK4 in triple-negative breast cancer.
I believe that the authors of the article cited in the prepared manuscript should be informed that a commentary on their previous research related to MAPK4 is being prepared. So that the authors of the previous paper could also refer to this "Commentaty".
Author Response
We thank the reviewer for her/his comments.
1. Please prepare Figure B similarly to Figure A. So that you can compare the TPM in all tested cell lines.
Response: Figure 1, panel A refers to mRNA expression data measured by qPCR in the indicated cell lines. Figure 1, panel B shows mRNA expression data measured by RNA-sequencing analysis in the two TNBC cell lines Hs578T and MDA-MB-231. We don't have internal RNA-seq data for the other cell lines. However, Figure 1, panel F reports mRNA expression data for all cell lines extracted from the CCLE database.
2. I think it would be worthwhile to also present the data on gene expression (at the mRNA level) of MAPK4 and MAPK6 in triple-negative breast cancer contained in the TCGA database.
Response: In this manuscript, we do not challenge the observation that MAPK4 gene is expressed in some TNBC patients. We question and challenge the data of Wang et al. showing that ERK4 is expressed in the TNBC cell lines Hs578T, MDA-MB-231 and HCC1937 and the consequent validation of the oncogenic role of ERK4 in these cellular models. We prefer not to show any data on MAPK4 or MAPK6 expression from TCGA patients since it may bring confusion to the main message of our manuscript.
3. I believe that the authors of the article cited in the prepared manuscript should be informed that a commentary on their previous research related to MAPK4 is being prepared. So that the authors of the previous paper could also refer to this "Commentary".
Response: The authors have been informed.